# Analysis of Kinematic Characteristics of Saanen Goat Spine under Multi-Slope

**DOI:** 10.3390/biomimetics7040181

**Published:** 2022-10-28

**Authors:** Fu Zhang, Xiahua Cui, Shunqing Wang, Haoxuan Sun, Jiajia Wang, Xinyue Wang, Sanling Fu, Zhijun Guo

**Affiliations:** 1College of Agricultural Equipment Engineering, Henan University of Science and Technology, Luoyang 471003, China; 2Henan International Joint Laboratory of Intelligent Agricultural Equipment Technology, Luoyang 471003, China; 3Collaborative Innovation Center of Machinery Equipment Advanced Manufacturing of Henan Province, Luoyang 471003, China; 4College of Physical Engineering, Henan University of Science and Technology, Luoyang 471003, China; 5College of Vehicle &Transportation Engineering, Henan University of Science and Technology, Luoyang 471003, China

**Keywords:** linear time normalization, quadruped robot, coupling motion, rigid and flexible coupling structure, Gaussian mixture model clustering algorithm, the kinematic characteristics of the spine

## Abstract

In order to improve the slope movement stability and flexibility of quadruped robot, a theoretical design method of a flexible spine of a robot that was based on bionics was proposed. The kinematic characteristics of the spine were analyzed under different slopes with a Saanen goat as the research object. A Qualisys track manager (QTM) gait analysis system was used to obtain the trunk movement of goats under multiple slopes, and linear time normalization (LTN) was used to calibrate and match typical gait cycles to characterize the goat locomotion gait under slopes. Firstly, the spatial angle changes of cervical thoracic vertebrae, thoracolumbar vertebrae, and lumbar vertebrae were compared and analyzed under 0°, 5°, 10°, and 15° slopes, and it was found that the rigid and flexible coupling structure between the thoraco–lumbar vertebrae played an obvious role when moving on the slope. Moreover, with the increase in slope, the movement of the spine changed to the coupling movement of thoraco–lumbar coordination movement and a flexible swing of lumbar vertebrae. Then, the Gaussian mixture model (GMM) clustering algorithm was used to analyze the changes of the thoraco–lumbar vertebrae and lumbar vertebrae in different directions. Combined with anatomical knowledge, it was found that the motion of the thoraco–lumbar vertebrae and lumbar vertebrae in the goat was mainly manifested as a left–right swing in the coronal plane. Finally, on the basis of the analysis of the maximin and variation range of the thoraco–lumbar vertebrae and lumbar vertebrae in the coronal plane, it was found that the coupling motion of the thoraco–lumbar cooperative motion and flexible swing of the lumbar vertebrae at the slope of 10° had the most significant effect on the motion stability. SSE, R^2^, adjusted-R^2^, and RMSE were used as evaluation indexes, and the general equations of the spatial fitting curve of the goat spine were obtained by curve fitting of Matlab software. Finally, Origin software was used to obtain the optimal fitting spatial equations under eight movements of the goat spine with SSE and adjusted-R^2^ as indexes. The research will provide an idea for the bionic spine design with variable stiffness and multi-direction flexible bending, as well as a theoretical reference for the torso design of a bionic quadruped robot.

## 1. Introduction

At present, most quadruped robots adopt rigid body structure, which is difficult to meet the needs of flexibility and bending when moving [1,2]. The flexible spine structure of tetrapods provides auxiliary power when they move so that they can show a variety of dexterous movement postures, demonstrating strong flexibility, stability, and environmental adaptability [3,4]. Hildebrand [5] studied the running process of cheetahs and horses, found that the torso of cheetahs was more curled and extended than that of horses, as well as the fact that the spine movement mainly acted on the legs, which could effectively increase the leg extension. They discussed the influence of leg movement on the running speed under the action of the spine. Hackert [6] observed the spinal movement of the pika. It was found that the spinal movement kept the body center of mass in a smooth motion trajectory that could effectively maintain the body balance and increase motion stability of the pika. Schilling et al. [7] used high-speed cineradiography to study the spinal intervertebral joint movement of mammals during running and semi-jumping, and it was found that spinal movement had an important impact on animal motor performance.

Therefore, the design of multidirectional flexible bending biomimetic spine structure, which is based on the motion characteristics of a tetrapod spine, is an important research direction in order to improve the kinematic capability of quadruped robots in an unstructured environment [8,9]. Zhang et al. [10] studied the spine structure and trunk movement characteristics of cheetahs and designed the passive compliant spine of a quadruped robot with adjustable stiffness on the basis of the bionics principle. It was found that when the spine stiffness was 681 N/m, the stable motion speed of the designed robot was the highest, which was 0.92 m/s. Zhao [11] designed and optimized the biomimetic trunk mechanism on the basis of the study of kinematic characteristics of goat torso and simulated it. It was found that the combination of goat kinematic characteristic parameters and robot motion control could effectively improve the robot motion effect. Wei [12] built a quadruped dynamic model to describe Transverse Gallop gait and Rotary Gallop gait. Through a large number of simulation experiments, it was found that the Transverse Gallop gait was designed for efficient, long-distance running, while the Rotary Gallop gait was designed for high-speed running in small-to-medium quadruped mobility systems. On the basis of Rotary Gallop gait, SQbot, a quadruped robot with bionic spinal joints, was developed. The overall trunk motion amplitude of the quadruped robot at higher moving speed was significantly reduced compared with that at lower moving speed. Li [13] studied the hopping gait of a quadruped robot with spinal joints and analyzed its gait parameters. Finally, a simplified model of a quadruped robot with spinal joints was established according to the bionics principle and the skeleton morphology of a cheetah. Physical validation found that the impedance coefficient of the robot with the best spine stiffness was 40% lower than that of the robot without the flexible spine. Borisova et al. [14] studied a method to match the dynamic model parameters of a cheetah with a running robot with a flexible spine through video tracking analysis of cheetah movement, and it was found that a flexible spine is the main feature of an efficient running robot.

Compared with other tetrapods, goats have smaller bodies, solid and powerful limb bones, and agile and light gait, and as well as this, they can adapt to different rugged terrain and road surface changes, showing strong movement stability [15,16,17]. In order to explore the structural characteristics of the goat spine, Van Engelen et al. [18] used a structural vibration test method to study the mechanical properties of goat spine motion segments. It was found that there were obvious formants in flexion and extension, lateral bending, and axial rotation of all goat lumbar spine motion segments, which verified the feasibility of the vibration test to evaluate the modal parameters of lumbar spine motion segments. Zhao [11] studied the trajectory changes of the trunk and the sensitivity of different parts by applying different load pressures on the trunk bones of goats. The difference in living environment between goats and sheep leads to the difference of their locomotor characteristics, but their spinal structures are the same, and thus the study of sheep static structural characteristics is of significance. Valentin et al. [19] found that the range of movement of the spine of sheep was smaller than that of humans, no matter what posture they adopted in comparison with the range of movement of the spine of people of different ages and sheep. DeVries et al. [20] tested the flexion and extension, lateral bending, and axial rotation of the cervical vertebrae of adult sheep (C2-C7) by a test device. They found the main behaviors and coupling behaviors of the C2-C7 segments of the cervical vertebrae of sheep. McDonald et al. [21] implanted steel beads into the vertebrae of sheep specimens, manually stretching neck extension and axial rotation. The three-dimensional position and direction of each cervical spine were determined using biplanar X-ray images. They verified the accuracy of model-based tracking measurement of three-dimensional dynamic cervical spine kinematics. At present, studies on the biological characteristics of goat spine are mainly focused on the analysis of the structural characteristics and mechanical properties of the dissected spine using specific methods, as well as the motion characteristics of goat spine under different states using a high-speed camera. However, there are few studies on spatial motion characteristics of goat spine through intra-anatomical spine motion under multiple slopes. Moreover, few scholars have established spinal spatial curves through spinal motion characteristics.

To explore the dynamic motion characteristics of goat spine and construct a mathematical model of a spine curve, the QTM gait analysis system was used to obtain the trunk movement of goats under multiple slopes. Combined with typical gait cycle, the spatial dynamic motion characteristics of goat spine were analyzed. The Gaussian mixture clustering model was used to analyze the influence of the motion characteristics of the spine in the anatomical plane on its spatial motion characteristics. Moreover, the slope with a significant effect was analyzed according to the data variation range of the spine in the anatomical plane under multiple slopes. The research will provide an idea for the bionic spine design with variable stiffness levels and multi-direction flexible bending, as well as providing a theoretical reference for the torso design of a bionic quadruped robot.

## 2. Materials and Methods

The Saanen goat was taken as the research object. A young male goat with moderate body size and good body condition was selected for the study. Before the experiment, the weight of the goat was measured at 19.50 kg by a scale (accuracy 0.01 kg), and the goat was scanned by CT tomography. During the scanning, the voltage was set to 140 kV, the current was 288 mA, and the layer thickness was 0.6 mm. The goat was anesthetized and asleep. The Dicom format file of the goat trunk was generated by CT scanning, and the three-dimensional structure of spine was reconstructed by Mimics software (Version 19.0, Materialize, Leuven, Belgium).

The spine of the selected subject included 7 cervical vertebrae (purple part), 13 thoracic vertebrae (green part), and 6 lumbar vertebrae (blue part) without obvious lesions. The seventh cervical vertebra showed obvious movement, and therefore the cervical markers were C at the first cervical vertebra and T at the seventh cervical vertebra. The thoracic vertebrae are connected to the ribs, which to some extent limit the excessive bending of the thoracic vertebrae, and the stability of the thoracic vertebrae is significantly higher than that of other spinal regions. Therefore, the markers for thoracic vertebrae were L1 at the 13th thoracic vertebrae, and L2 at the third lumbar vertebrae and L3 at the sixth lumbar vertebrae were selected as markers for lumbar vertebrae. Hair on the spine of the test subject was removed before labeling. Reflective balls were pasted at T (the most obvious position at the junction between cervical vertebrae and thoracic vertebrae), L1 (the thoracic vertebrae connected with the last rib of the goat), L2 (the lumbar vertebrae connected with the hip bone of the goat), and L2 (the middle position of L1 and L3). In addition, reflective balls were pasted on the limbs to obtain the locomotion gait of the goat, and LFn (left front), LHn (left hind), RFn (right front), and RHn (right hind) were labeled. The attachment position of the reflective balls is shown in Figure 1. In this paper, the change of angle CTL1 represented the motion of cervical vertebrae relative to thoracic vertebrae of goats, the change of angle TL1L2 represented the motion of thoracic vertebrae relative to lumbar vertebrae of goats, and the change of angle L1L2L3 represented the internal motion of lumbar vertebrae of goats. They were labeled as *θ*_1_, *θ*_2_, and *θ*_3_, respectively, for subsequent analysis.

A test platform consisting of a 6-DOF motion platform, a motion capture system (Qualisys Miqus M3 model, displacement error ≤ 0.1 mm, frame rate of 340 fps, system delay ≤ 5 ms), and a data acquisition and analysis system was used for the test. This is shown in Figure 2. Before the experiment, the platform was calibrated, and the basic space coordinate system was established. The positive direction of the X-axis was the movement direction of the goat, the positive direction of the Y-axis was the horizontal direction of the left side of the advance, and the positive direction of the Z-axis was the vertical geodetic direction. The sampling frequency of the motion capture system was set to 100 Hz, and the frame rate was set to 100 fps. The angle between the platform and the Z-axis is α. The slope test was carried out by changing the angle value of α, and the inclination angles α were 0°, 5°, 10°, and 15°. During the experiment, the goat’s horns were tied with a rope and pulled by the experimenter at the back end. The rope was relaxed to ensure that the goat head was in a state of free movement, and the goat was lured forward by food induction.

It was found that goat mainly adopted a “three-phase supported, two-phase supported, three-phase supported” gait when moving on the slope. That is, when starting, one front leg swings and the other three legs support. Before the swing phase falls to the ground and becomes the supported phase, the other leg is raised and swings. In this study, the leg lifting sequence of “left front–right hind–right front–left hind” was analyzed as a typical gait cycle, and the locomotion gait is shown in Figure 3, which consisted of eight movements (①–⑨, with the ⑨ movement being the same as the ① movement).

In order to accurately screen the gait cycle, the movement track of the limb markers was observed. The movement track of the third limb marker along the positive direction of X-, Y-, and Z-axes under the slope of 0° is shown in Figure 4.

It is easy to see that the X and Z direction trajectories of the marker points had obvious periodicity, so when
(1)$$\left|X_{2}-X_{1}\right|<3\;\mathrm{mm},$$
(2)$$\left|Z_{2}-Z_{1}\right|<1\;\mathrm{mm},$$
the leg was considered to be in the supported posture and it was the supported phase.

In order to study the kinematic characteristics of the goat spine at different slopes, the key roles of each part of the spine were explored through the spatial curvature of the spine and the maximin and variation range of *θ*_1_, *θ*_2_, and *θ*_3_. Then Gaussian mixture model (GMM) clustering method was used to determine the main direction of spinal curvature. Moreover, the main acting slope of the goat spine was analyzed, combined with the maximin and variation range of *θ*_2_ and *θ*_3_ in the main bending direction. Finally, the spine curve of eight movements under the main action slope was fitted.

## 3. Results

In order to describe the motion affected by the speed of goat movement, three groups of data were selected for analysis at different slopes to illustrate the general motion characteristics of the goat spine. For the same kind of movement, the goat movement speed and the step length were different, resulting in different gait cycle time. Linear time normalization (LTN) was used for sequence calibration and matching of gait cycle. The normalization mapped the data to the interval [0, 1] without changing the data features, so as to facilitate the comparison and analysis of spine movement during gait cycle. The normalization formula is as follows:(3)$${\mathrm{Frame}}^{\prime }=\frac{{\mathrm{Frame}}_{n}-{\mathrm{Frame}}_{\min}}{{\mathrm{Frame}}_{\max}-{\mathrm{Frame}}_{\min}}$$
where Frame_min_ is the minimum frame in the gait cycle and Frame_max_ is the maximum frame in the gait cycle. Figure 5, Figure 6, Figure 7 and Figure 8 were drawn with the normalized gait cycle from ① movement to ⑧ movement as the horizontal axis and the change of *θ*_1_, *θ*_2_, and *θ*_3_ as the vertical axis.

### 3.1. Analysis of the Motion of Each Part of Spine

Combined with the eight movements under the 0° slope as shown in Figure 5d (see Figure 3 for the explanation of movements), the general changes of *θ*_1_, *θ*_2_, and *θ*_3_ as shown in Figure 5a–c were analyzed. It was found that the overall change trend of all angles was the same. When the front limb swung, the angle *θ*_1_ decreased, and when the hind limb swung, the angle *θ*_1_ increased. When the front limb was supported, the angle *θ*_1_ reached the peak, and when the hind limb was supported, the angle *θ*_1_ reached the minimum. When the right front leg swung, the angle of *θ*_2_ reached the peak value of 170°. When the motion state of the goat ipsilateral leg switched, the angle change rate of *θ*_2_ decreased. The angle change rate of *θ*_2_ increased when the motion state of the corresponding leg switched. When the left front leg swung monophasically, the angle of *θ*_3_ increased slowly. When the left front leg and right hind leg swung, the angle change rate of *θ*_3_ increased. When the right front leg swung monophasically, *θ*_3_ reached the peak and then decreased, and the peak value was about 170°. When the left hind leg swung monophasically, the angle *θ*_3_ decreased.

**Figure 5 biomimetics-07-00181-f005:**
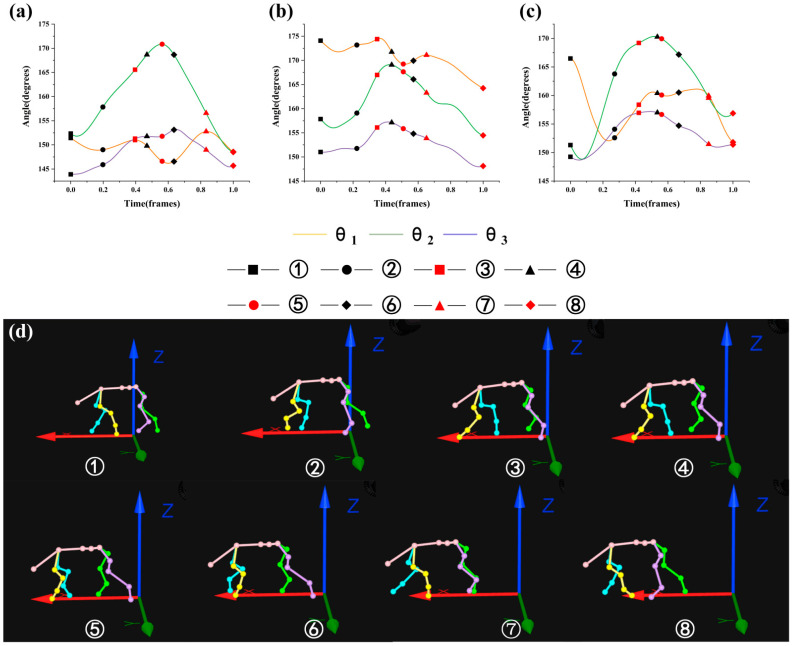
Motion state of goat: (**a**–**c**) are angle changes of *θ*_1_, *θ*_2_, and *θ*_3_ under the slope of 0°; (**d**) eight movements under the slope of 0°.

Combined with the eight movements under the 5° slope as shown in Figure 6d (see Figure 3 for the explanation of movements), the general changes of *θ*_1_, *θ*_2_, and *θ*_3_ as shown in Figure 6a–c were analyzed. It was found that when the left front leg swung, the angle of *θ*_1_ changed about 5°, the angle of *θ*_2_ decreased, and the angle of *θ*_3_ decreased. When the left front leg and right hind leg swung, the angle of *θ*_1_ increased, the angle of *θ*_2_ decreased, and the angle of *θ*_3_ changed about 5°. When the right hind leg swung monophasically, the angle of *θ*_1_ changed about 3°, the angle of *θ*_2_ changed about 3°, and the angle of *θ*_3_ changed about 4°. When the right front leg and right hind leg swung on the same side, the angle of *θ*_1_ decreased, the angle of *θ*_2_ changed about 3°, and the angle of *θ*_3_ changed about 5°. When the right front leg swung monophasically, the angle of *θ*_1_ changed about 5°, the angle of *θ*_2_ increased, and the angle of *θ*_3_ changed about 5°. When the right front leg and left hind leg swung on the different side, the angle of *θ*_1_ and *θ*_2_ increased, while the angle of *θ*_3_ decreased. When the left hind leg swung monophasically, the angle of *θ*_1_, *θ*_2_, and *θ*_3_ all decreased.

**Figure 6 biomimetics-07-00181-f006:**
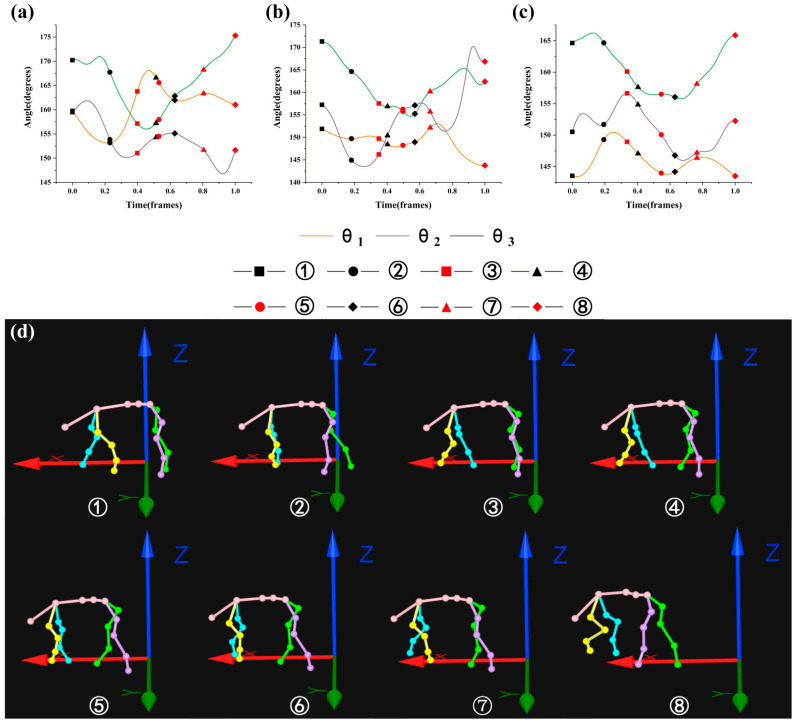
Motion state of goat: (**a**–**c**) are angle changes of *θ*_1_, *θ*_2_, and *θ*_3_ under the slope of 5°; (**d**) eight movements under the slope of 5°.

Combined with the eight movements under 10° slope as shown in Figure 7d (see Figure 3 for the explanation of movements), the general changes of *θ*_1_, *θ*_2_, and *θ*_3_ as shown in Figure 7a–c were analyzed. It was found that the angle of *θ*_1_ and *θ*_3_ decreased, and the angle of *θ*_2_ changed about 10° when the left front leg swung. When the left front leg and right hind leg swung, the angle of *θ*_1_ and *θ*_3_ changed about 5°, and the angle of *θ*_3_ increased. When the right hind leg swung, the angle of *θ*_1_ changed about 2°, the angle of *θ*_2_ changed about 5°, and the angle of *θ*_3_ increased. When the right front and right hind legs swung on the same side, the angle of *θ*_1_ changed about 2°, the angle of *θ*_2_ decreased, and the angle of *θ*_3_ changed about 10°. When the right front leg swung monophasically, the angle of *θ*_1_ changed about 2°, and the angle of *θ*_2_ and *θ*_3_ increased. When the right front leg and left hind leg swung on the different side, the angle of *θ*_1_ decreased, the angle of *θ*_2_ increased, and the angle of *θ*_3_ increased after decreasing. When the left hind leg swung monophasically, the angle of *θ*_1_ decreased, the angle of *θ*_2_ basically stayed the same, and the angle of *θ*_3_ decreased by about 5°.

**Figure 7 biomimetics-07-00181-f007:**
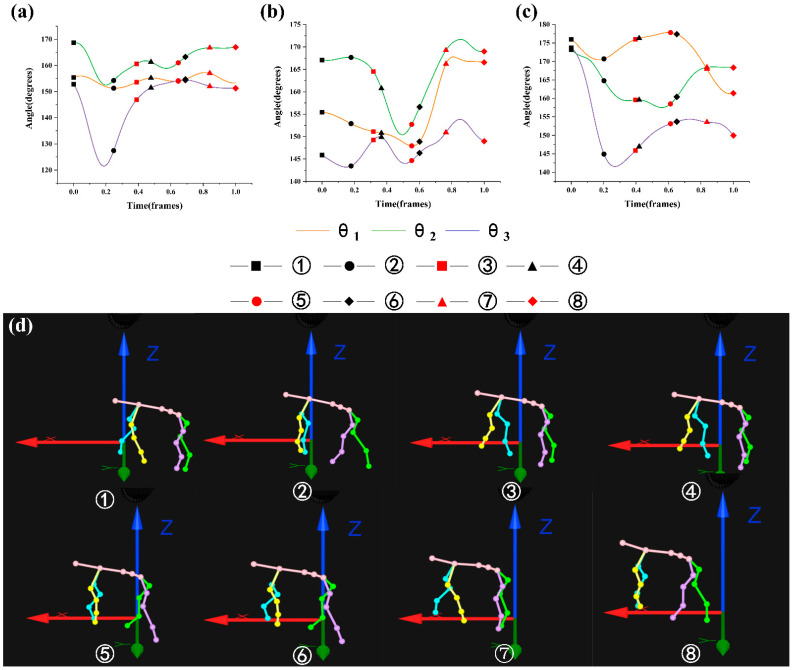
Motion state of goat: (**a**–**c**) are angle changes of *θ*_1_, *θ*_2_, and *θ*_3_ under the slope of 10°; (**d**) eight movements under the slope of 10°.

Combined with the eight movements under the 15° slope as shown in Figure 8d (see Figure 3 for the explanation of movements), the general changes of *θ*_1_, *θ*_2_, and *θ*_3_ as shown in Figure 8a–c were analyzed. It was found that the angle of *θ*_1_ decreased, the angle of *θ*_2_ increased, and the angle of *θ*_3_ changed to about 155° when the left front leg swung. When the left front leg and right hind leg swung on the different side, the angle of *θ*_1_ increased, the angle of *θ*_2_ changed about 3°, and the angle of *θ*_3_ decreased. When the right hind leg swung, the angle of *θ*_1_ changed about 5°, the angle of *θ*_2_ decreased, and the angle of *θ*_3_ rose about 1°. When the right front leg and right hind leg swung on the same side and the right front leg swung monophasically, the angle of *θ*_1_ decreased, and the angle of *θ*_2_ and *θ*_3_ increased. When the right front leg and left hind leg swung on the different side, the angle of *θ*_1_ increased, the angle of *θ*_2_ changed about 4°, and the angle of *θ*_3_ changed about 5°. When the left hind leg swung monophasically, the angle of *θ*_1_ decreased, the angle of *θ*_2_ increased, and the angle of *θ*_3_ increased after it decreased.

**Figure 8 biomimetics-07-00181-f008:**
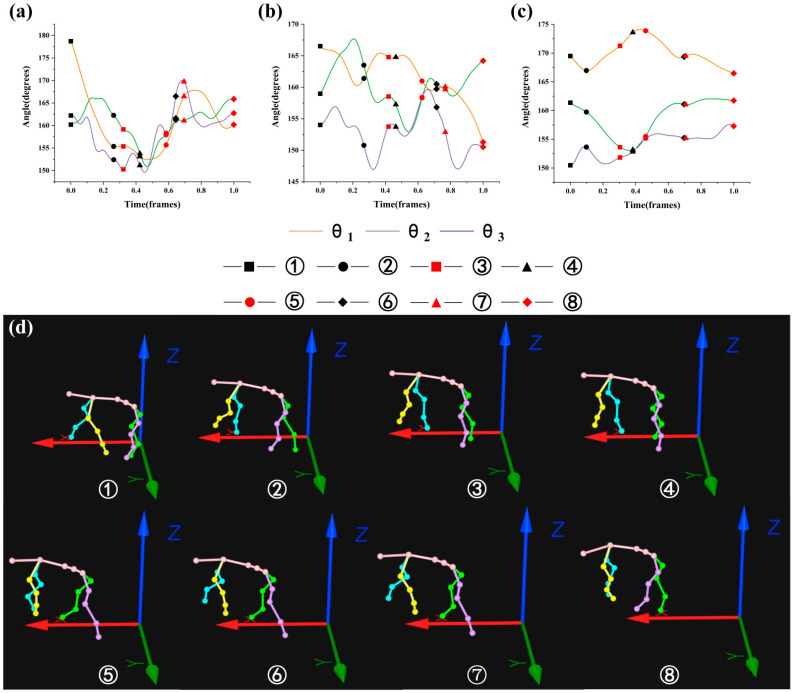
Motion state of goat: (**a**–**c**) are angle changes of *θ*_1_, *θ*_2_, and *θ*_3_ under the slope of 15°; (**d**) eight movements under the slope of 15°.

Through the above analysis, it was found that the leg movement of goats was closely related to the spine movement, and the spine movement showed different phenomena at different slopes. Therefore, the spine played a significant role in the movement process of goats under slopes.

### 3.2. Analysis of the Maximin and Variation Range of θ_1_, θ_2_, and θ_3_

The analysis of the maximin and range of the change of each key angle of the spine under different slopes is an important method in terms of studying the action of each part of the spine. The motion of the cervical thoracic vertebrae, thoraco-lumbar vertebrae, and lumbar vertebrae were analyzed by the maximin and variation range of the key spinal angles (*θ*_1_, *θ*_2_, and *θ*_3_). The maximin and variation range of them are shown in Figure 9.

According to Figure 9a, the angle of *θ*_1_ decreased first and then increased with the increase in the slope. At the same slope, *θ*_2_ was larger than *θ*_3_, indicating that the role of thoracolumbar coordination was more significant in the process of goat movement. Since the thoracic vertebrae has better stability than the lumbar vertebrae to some extent, the motion of the thoracolumbar vertebrae could be regarded as the motion of rigid and flexible coupling structure. Through the analysis shown Figure 9b, it was found that the variation range of *θ*_2_ was larger than that of *θ*_3_ under the slope of 0°. Conversely, the variation range of *θ*_2_ was smaller than that of *θ*_3_ under the slopes of 5°, 10°, and 15°, indicating that with the increase in slope, the regulation function of the lumbar spine was increasingly more important. The main movement of the spine changed to the coupling movement between the thoracolumbar coordination movement and the flexible swing of the lumbar vertebrae. The range of *θ*_1_ gradually increased with the increase in slope, indicating that the head had an important regulating effect on the movement stability of the goat under large slopes. This was consistent with the team’s previous research results [16].

### 3.3. Analysis of Motion of Thoracic and Lumbar Vertebrae under Different Slopes

The goat advanced along the positive direction of the X-axis. In order to further describe the motion of thoracolumbar vertebrae and lumbar vertebrae, the motion of *θ*_2_ and *θ*_3_ angles under different slopes was studied in combination with anatomical orientation terms. In the motion coordinate system, the motion of the spine was mainly represented by the motion (up and down fluctuation) in the XZ plane, which is the sagittal plane, and the motion (left and right swing) in the XY plane, which is the coronal plane.

The Gaussian mixture model (GMM) is a common data clustering method that combines similar data together according to the similarity [22,23]. On the basis of the GMM method and expected maximum (EM) algorithm, the effects of covariance matrix types and initial conditions on the clustering results of Gaussian mixture models was studied. The optimal combination of two covariance matrices of diagonal and full and the conditions of identical and non-identical covariance matrices was explored.

Firstly, the clustering model between the data of *θ*_2_ in the XZ and XY planes and the spatial data was established to analyze the influence of the motion of the thoracic–lumbar vertebra coupling structure in the two planes. The clustering results of θ_2_ data in XZ and XY planes are shown in Figure 10.

By comparing (a) with (e), (b) with (f), and (c) and (g) in Figure 10, it was found that under the same covariance matrix and initial conditions, the data clustering effect of *θ*_2_ in the XY plane was obvious and could be clearly divided into four independent and complete regions. That was, the data could be accurately divided into the data under four slopes. According to this, the motion of *θ*_2_ in the XY plane had the most obvious corresponding relationship with the slope. In this case, the motion of the thoracolumbar spine under different slopes was mainly represented by the left and right swing in the coronal plane.

As shown in Figure 10e–g, it was found that when the covariance matrix was diagonal and different covariance matrices were selected, the four slope regions were most clearly delimited, and there was a certain distance between the regions. This indicated that the GMM had the best clustering effect under this condition, and the angle data under different slopes accurately corresponded to the corresponding slopes, which could be used to analyze the main direction of spinal curvature. Then, under the condition, clustering models between the angle data of *θ*_3_ in XZ and XY planes and spatial data were established in order to analyze the influence of lumbar curvature in the sagittal plane and in the coronal plane on spinal motion. The clustering results of θ_3_ in the XZ and XY planes are shown in Figure 11.

As can be seen from Figure 11, the data clustering effect of the *θ*_3_ angle in the XZ plane was not obvious and could not be clearly divided into four independent and complete regions. However, the clustering effect of the data in the XY plane was obvious, that is, the data could be accurately divided according to four slopes. Therefore, it was concluded that the change of *θ*_3_ in the XY plane was more significant, that is, the motion of goat lumbar vertebrae was mainly manifested as left–right swing in the coronal plane.

### 3.4. Analysis of the Variation Range of θ_2_ and θ_3_ in the XY Plane

The motion of thoracic vertebrae and lumbar vertebrae was mainly manifested as left–right swing in the coronal plane, and the variation range of *θ*_2_ and *θ*_3_ reflected the influence of thoracic vertebrae and lumbar vertebrae on the body of goats at different slopes. The larger the range of variation, the greater the regulation effect of this part on the movement of goats. The variation range of *θ*_2_ and *θ*_3_ in the XY plane under different slopes is shown in Figure 12.

It can be seen from Figure 12 that the variation range of *θ*_2_ in the XY plane was essentially stable at 20–23°, while the variation range of *θ*_3_ in the XY plane varied greatly. As the slope increased to 10°, the variation range of *θ*_3_ in the XY plane increased gradually to 25°, and the variation range decreased when the slope increased to 15°. A comprehensive analysis of the variation range of *θ*_2_ and *θ*_3_ in the XY plane showed that the variation range of *θ*_2_ and *θ*_3_ in the XY plane was the largest at the slope of 10°, which meant that the coupling motion of thoraco-lumbar cooperative motion and flexible swing of lumbar vertebrae at the slope of 10° had the most significant regulating effect.

### 3.5. Construction of Spine Spatial Curve Models for Eight Movements at 10°

Matlab software was used to fit the spatial curve polynomial of the eight movements of the goat spine under 10° slope. The spinal space curve was generated by combining the two-dimensional fitting curves in the coronal plane and sagittal plane. The data of C, T, L1, L2, and L3 points (x, y, z) were imported into Matlab software, and the curve of the spine in the coronal plane was fitted with x as the independent variable and y as the dependent variable. Conversely, the curve of spine in sagittal plane was fitted with x as the independent variable and z as the dependent variable. Finally, the general equation of spine space fitting curve was obtained after analyzing the fitting results. The fitting results of different indices of x in the coronal plane and sagittal plane are shown in Table 1 and Table 2, respectively.

Because the fitting result of the spinal curve in the coronal plane was uncertain when the independent variable index was four, the fitting effect of the spinal curve was analyzed when the independent variable index increased from one to three. As can be seen from Table 1, with the increase in independent variable index, the SSE of the spine curve fitting in the coronal plane gradually decreased and the R^2^ increased. In addition, the influence of the number of independent variables included in the equation should be considered. The Adjusted-R^2^ of group 2 and 3 increased with the increase in independent variables. When n was 3, SSE were 159.53, 318.17, and 0.33; R^2^ were 0.9760, 0.9881, and 1.0000, Adjusted-R^2^ were 0.9041, 0.9524, and 0.9999; and RMSE were 12.631, 17.837, and 0.573 for groups 1, 2, and 3, respectively. It met the fitting requirements.

As can be seen from Table 2, with the increase in the independent variable index, SSE and RMSE decreased, while R^2^ and Adjusted-R^2^ constantly increased, indicating that with the increase in the index of x, the fitted curve was closer to the curve characteristics of the actual spine. When the independent variable index was 2 or 3, R^2^ was above 0.9885, and Adjusted-R^2^ was above 0.9770. When the index of x increased from 2 to 3, SSE decreased by 127.81, 55.23, and 33.44; R^2^ increased by 0.0111, 0.0049, and 0.0067; Adjusted-R^2^ increased by 0.0212, 0.0008, and 0.0124; and RMSE decreased by 5.876, 4.5178, and 3.8783 for groups 1, 2, and 3, respectively. However, with the increase in the index of independent variables, the complexity of the spinal space curve increased. When n was 3, SSE were 159.53, 318.17, and 0.33; R2 were 0.9760, 0.9881, and 1.0000; Adjusted-R2 were 0.9041, 0.9524, and 0.9999; and RMSE were 12.631, 17.837, and 0.573 for groups 1, 2, and 3, respectively. It met the fitting requirements.

Therefore, the general equation of fitting curve of goat spine in coronal plane was
(4)$$y=f\left(x\right)={\mathrm{ax}}^{3}+{\mathrm{bx}}^{2}+\mathrm{cx}+d$$
the general equation of fitting curve in sagittal plane was
(5)$$z=f\left(x\right)={\mathrm{ex}}^{2}+\mathrm{fx}+g$$
the spatial fitting curve equations of goat spine could be expressed as
(6)$$\{\begin{array}{c} y=f\left(x\right)=ax^{3}+bx^{2}+cx+d\\ z=f\left(x\right)=ex^{2}+fx+g \end{array}$$

The independent variable x, dependent variables y and z, parameter names, and functions were set through the function editor function of Origin software, and the setup function was compiled. The nonlinear fitting function was used to fit the spine curve, and the equation parameters A, B, C, D, E, F, and G were obtained. As three groups of data were selected for fitting, the curve equations with the best fitting accuracy were selected to represent the spine curves. The fitting results under the eight movements are shown in Table 3.

As can be seen from Table 3, the Adjusted-R^2^ of the second group of data was smaller than that of the first and third groups, while the SSE was larger, indicating that the fitting effect of the second group of data was the worst. A comprehensive comparison of the Adjusted-R^2^ and SSE of the first group and the third group showed that the Adjusted-R^2^ of the two groups were above 0.9888. The SSE of the first group of ①–⑥ movements were higher than that of the third group, and the SSE of the first group of ⑦ and ⑧ movements were lower than that of the third group. Therefore, the third set of fitting equations was chosen to represent the spinal space curves. The specific curve equations of the eight movements were as follows:

① The left front leg swung monophasically,
(7)$$\left\{ \begin{array}{c} y=f\left(x\right)=-9.3346e^{-7}x^{3}-5.13986e^{-4}x^{2}-0.29539x+189.63352\\ z=f\left(x\right)=-2.38194e^{-4}x^{2}+0.06525x+421.60546 \end{array}\right.$$

② The left front leg and the right hind leg swung biphasically,
(8)$$\left\{ \begin{array}{c} y=f\left(x\right)=-2.2686e^{-6}x^{3}-3.993e^{-4}x^{2}-0.05148x+215.77565\\ z=f\left(x\right)=-2.49836e^{-4}x^{2}+0.18044x+438.47413 \end{array}\right.$$

③ The right hind leg swung monophasically,
(9)$$\left\{ \begin{array}{c} y=f\left(x\right)=-1.1923e^{-6}x^{3}+2.00709e^{-4}x^{2}-0.10865x+223.85696\\ z=f\left(x\right)=-1.92079e^{-4}x^{2}+0.19724x+442.01349 \end{array}\right.$$

④ The right hind leg and the right front leg swung biphasically,
(10)$$\left\{ \begin{array}{c} y=f\left(x\right)=-1.10815e^{-6}x^{3}+2.34316e^{-4}x^{2}-0.12049x+225.50681\\ z=f\left(x\right)=-1.84803e^{-4}x^{2}+0.19497x+443.03394 \end{array}\right.$$

⑤ The right front leg swung monophasically,
(11)$$\left\{ \begin{array}{c} y=f\left(x\right)=-3.494e^{-7}x^{3}+1.10674e^{-4}x^{2}-0.13416x+209.40728\\ z=f\left(x\right)=-1.57941e^{-4}x^{2}+0.24015x+438.63463 \end{array}\right.$$

⑥ The right front leg and the left hind leg swung biphasically,
(12)$$\left\{ \begin{array}{c} y=f\left(x\right)=-5.03574e^{-7}x^{3}+3.00916e^{-4}x^{2}-0.17334x+201.31853\\ z=f\left(x\right)=-1.39878e^{-4}x^{2}+0.24153x+436.80313 \end{array}\right.$$

⑦ The left hind leg swung monophasically,
(13)$$\left\{ \begin{array}{c} y=f\left(x\right)=-2.17272e^{-6}x^{3}+0.00262x^{2}-1.02443x+254.14109\\ z=f\left(x\right)=-2.07211e^{-5}x^{2}+0.16972x+451.53693 \end{array}\right.$$

⑧ The left hind leg and the left front leg swung biphasically,
(14)$$\left\{ \begin{array}{c} y=f\left(x\right)=-3.21143e^{-6}x^{3}+0.00492x^{2}-2.42112x+496.72515\\ z=f\left(x\right)=-1.88935e^{-5}x^{2}+0.22273x+432.31501 \end{array}\right.$$

## 4. Discussion

In this study, the QTM gait analysis system was used to analyze the motion characteristics of the spine at different slopes in combination with the typical gait cycle of the goat. Compared with the traditional spine biological characteristics analysis method, it can obtain the changes of goat spine under real movement state, which provides a reference for the subsequent study of goat movement mechanism.

(1) The change of CTL1 represented the motion of cervical vertebrae relative to thoracic vertebrae, the change of TL1L2 represented the motion of thoracic vertebrae relative to lumbar vertebrae, and the change of L1L2L3 represented the internal motion of lumbar vertebrae, which were labeled as *θ*_1_, *θ*_2_, and *θ*_3_, respectively, for the convenience of subsequent research. On the basis of LTN, the sequence calibration and matching were performed. Moreover, the “left front–right hind–right front–left hind” leg lifting sequence was used as a typical gait cycle to analyze the changes of *θ*_1_, *θ*_2_, and *θ*_3_ under different slopes. It was found that the changes of *θ*_1_, *θ*_2_, and *θ*_3_ of the eight movements had a general law, which indicated that the spine motion characteristics were significant in the motion process of the goat under different slopes. It had a regulating effect on the stability of the goat motion.

(2) The maximin and variation range of *θ*_1_, *θ*_2_, and *θ*_3_ under 0°, 5°, 10°, and 15° slopes were analyzed and compared. It was found that the maximin of *θ*_2_ was greater than that of *θ*_3_ at different slopes, and the variation range of *θ*_2_ changed from larger than that of *θ*_3_ to smaller than that of *θ*_3_ with the increase in slope. That is to say, lumbar vertebrae played an increasingly important role in regulation. Spinal movement under large slopes was mainly manifested as a coupled motion of thoraco–lumbar coordination motion and flexible swing of lumbar vertebrae. Moreover, the head played an important role in regulating the motion stability of the goat under large slopes.

(3) According to the similarity between the data of anatomic plane and the spatial data, the Gaussian mixture model clustering algorithm was used to analyze the spatial variation of *θ*_2_ in thoracolumbar vertebrae and *θ*_3_ in a lumbar intervertebral body. It was found that when the covariance matrix was diagonal and the covariance matrix was different, the clustering effect of angle variation data under different slopes was obvious. The motion of thoracolumbar vertebrae and lumbar vertebrae was mainly manifested as left–right swing in the coronal plane.

(4) According to the range of data variation, the regulation effect of each part of the spine on goat movement was studied, and the range of angle variation in the coronal plane of *θ*_2_ and *θ*_3_ was comprehensively analyzed. It was found that the variation range of *θ*_2_ in the coronal plane was essentially stable at 20–23° when the slope changed from 0° to 15°, and the variation range of *θ*_3_ in the coronal plane was about 25° at 10°. The results showed that the coupling motion of thoracolumbar joint motion and flexible swing of lumbar vertebrae at 10° had the most significant effect on the kinematic stability of the goat.

The previous research of the team found that goats have excellent adhesion performance and can move stably in the slope environment. At present, only the motion characteristics of the spine under 0°, 5°, 10°, and 15° slopes in combination with the structural characteristics of the goat spine were analyzed in this study. Moreover, combined with the motion characteristics of the spine, the optimal fitting space equations were established under the eight movements. However, the best fitting curve based on the adhesion coefficient judgment still needs to be determined to further design the multidirectional flexible bending bionic spine. Limited by the experimental conditions, only laboratory slope tests have been conducted to obtain the laboratory data of goat climbing movement. However, the characteristics of goat spine movement under a real slope environment and larger slopes need to be further studied. The excellent locomotor stability of the goat is the coordinated action of spine and limbs. The further interaction mechanism of the spine and limbs remains to be studied. In future research, we will use high-speed camera technology to study the movement state of goats in a real environment and analyze the coordination between the spine and limbs on the basis of image research.

## Figures and Tables

**Figure 1 biomimetics-07-00181-f001:**
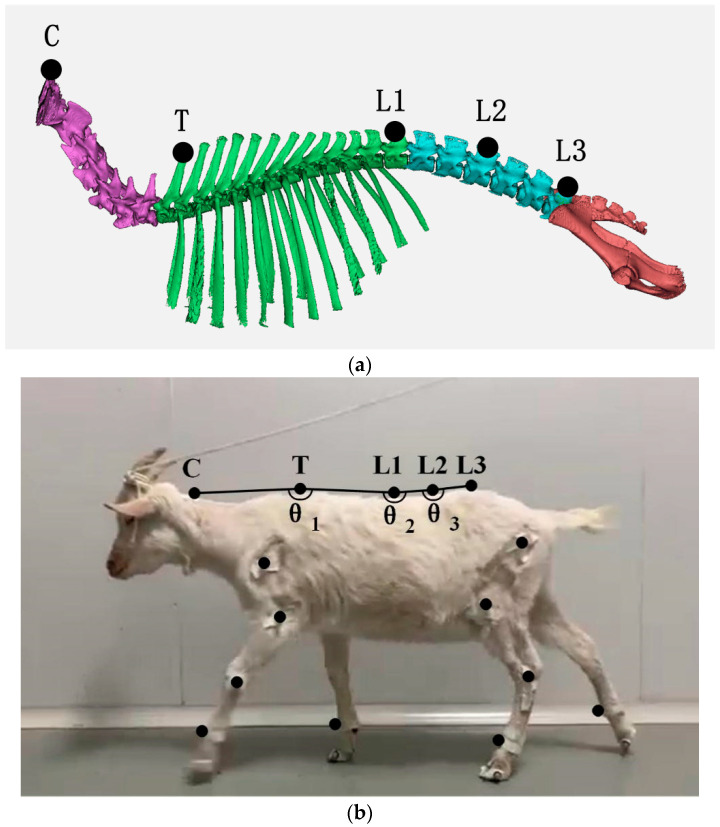
Placement map of reflective balls pasted on the spine: (**a**) schematic representation of the position of spinal markers; (**b**) location map of spinal markers for the test subject.

**Figure 2 biomimetics-07-00181-f002:**
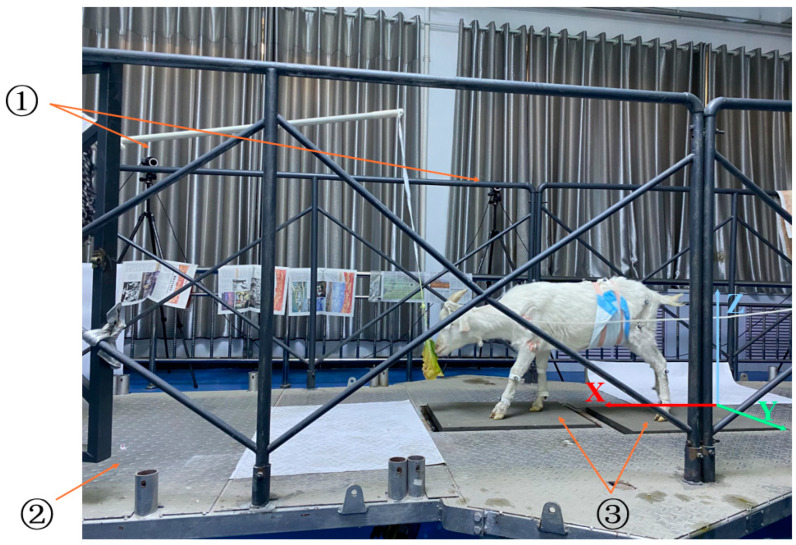
The motion test platform consists of ① the motion capture system, ② 6-DoF motion platform, and ③ the three-dimensional biomechanical force measuring table.

**Figure 3 biomimetics-07-00181-f003:**
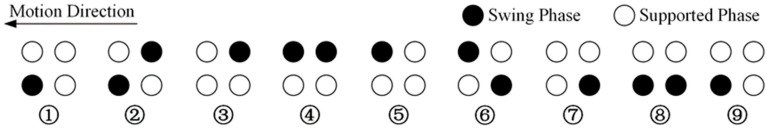
Goat locomotion gait on the slopes consisting of eight movements, which are ① the left front leg swung monophasically; ② the left front leg and the right hind leg swung biphasically; ③ the right hind leg swung monophasically; ④ the right hind leg and the right front leg swung biphasically; ⑤ the right front leg swung monophasically; ⑥ the right front leg and the left hind leg swung biphasically; ⑦ the left hind leg swung monophasically; ⑧ the left hind leg and the left front leg swung biphasically.

**Figure 4 biomimetics-07-00181-f004:**
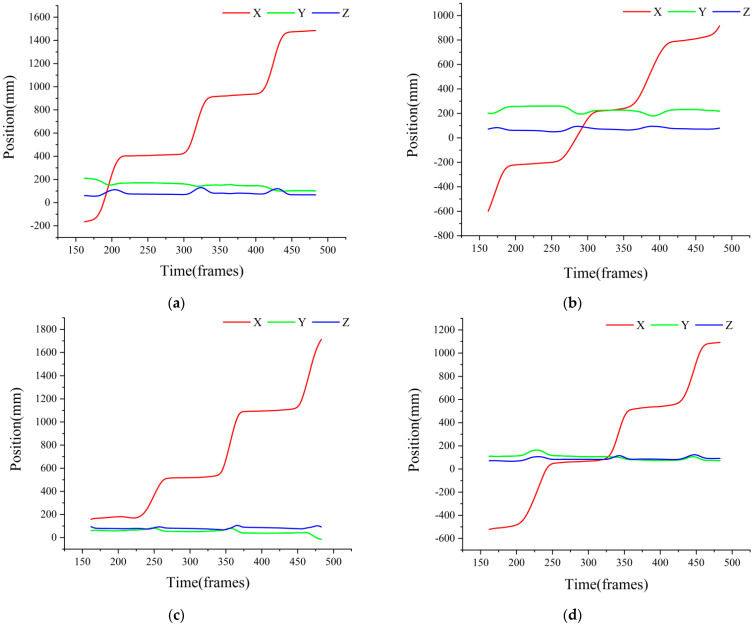
The trajectory of the third marker of the limbs along the X-, Y-, and Z-directions under the slope of 0°: (**a**) the trajectory of the third marker of the left front leg; (**b**) the trajectory of the third marker of the left hind leg; (**c**) the trajectory of the third marker of the right front leg; (**d**) the trajectory of the third marker of the right hind leg.

**Figure 9 biomimetics-07-00181-f009:**
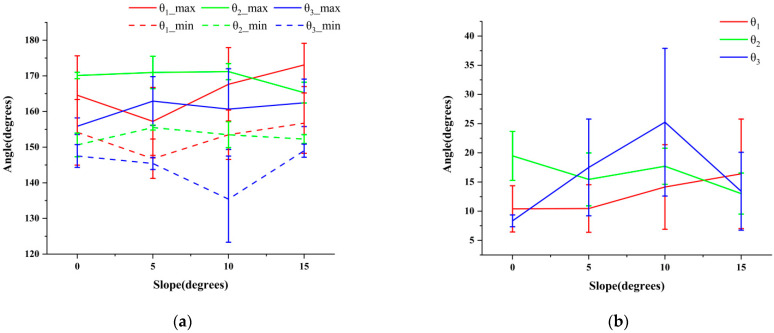
The maximin and variation range of *θ*_1_, *θ*_2_, and *θ*_3_: (**a**) maximum value and minimum value; (**b**) variation range.

**Figure 10 biomimetics-07-00181-f010:**
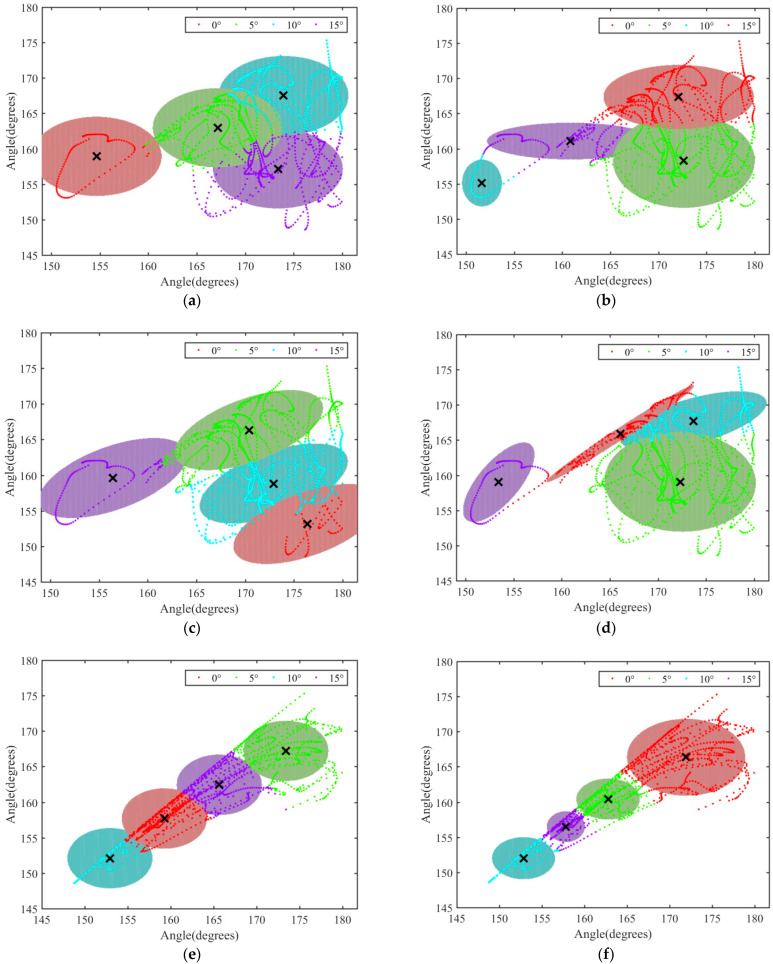
Clustering results of *θ*_2_ data in the XZ and XY planes: (**a**) angle data of *θ*_2_ in the XZ plane, when Sigmal was diagonal, and shared covariance was true; (**b**) angle data of *θ*_2_ in the XZ plane, when Sigmal was diagonal, and shared covariance was false; (**c**) angle data of *θ*_2_ in the XZ plane, when Sigmal was full, and shared covariance was true; (**d**) angle data of *θ*_2_ in the XZ plane, when Sigmal was full, and shared covariance was false; (**e**) angle data of *θ*_2_ in the XY plane, when Sigmal was diagonal, and shared covariance was true; (**f**) angle data of *θ*_2_ in the XY plane, when Sigmal was diagonal, and shared covariance was false; (**g**) angle data of *θ*_2_ in the XY plane, when Sigmal was full, and shared covariance was true; (**h**) angle data of *θ*_2_ in the XY plane, when Sigmal was full, and shared covariance was false.

**Figure 11 biomimetics-07-00181-f011:**
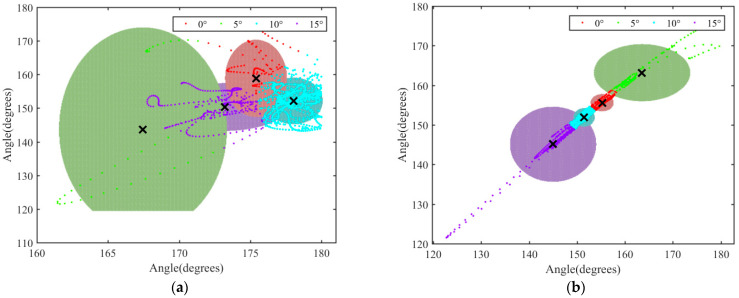
Clustering results of *θ*_3_ data in the XZ and XY planes: (**a**) angle data of *θ*_3_ in the XZ plane, when Sigmal was diagonal, and shared covariance was false; (**b**) angle data of *θ*_3_ in the XY plane, when Sigmal was diagonal, and shared covariance was false.

**Figure 12 biomimetics-07-00181-f012:**
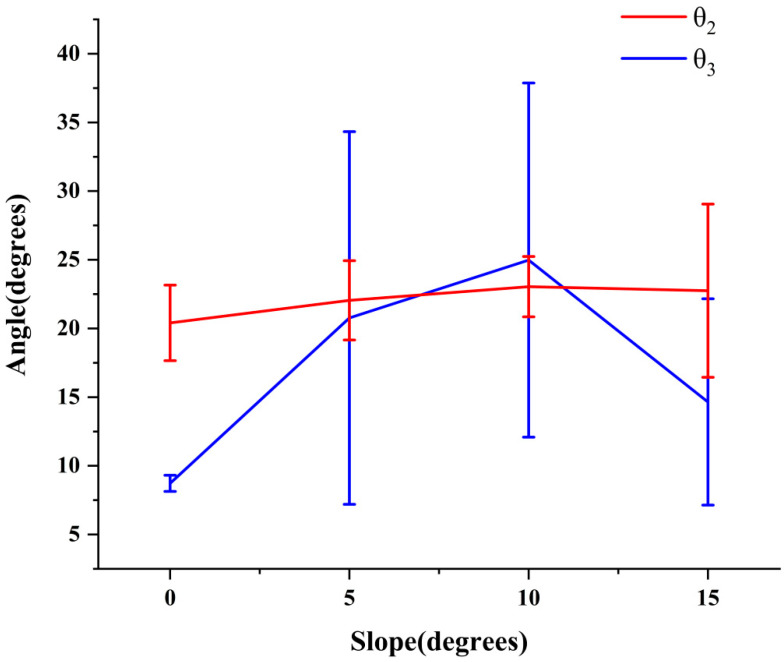
The variation range of *θ*_2_ and *θ*_3_.

**Table 1 biomimetics-07-00181-t001:** Fitting results of different indices of x in the coronal plane.

Group	x^n^	SSE	R^2^	Adjusted-R^2^	RMSE
1	x	299.67	0.9550	0.9399	9.995
	x^2^	297.03	0.9553	0.9107	12.187
	x^3^	159.53	0.9760	0.9041	12.631
2	x	1308.00	0.9511	0.9348	20.880
	x^2^	715.62	0.9732	0.9465	18.916
	x^3^	318.17	0.9881	0.9524	17.837
3	x	70.32	0.9946	0.9928	4.842
	x^2^	66.71	0.9949	0.9898	5.775
	x^3^	0.33	1.0000	0.9999	0.573

**Table 2 biomimetics-07-00181-t002:** Fitting results of different indices of x in the sagittal plane.

Group	x^n^	SSE	R^2^	Adjusted-R^2^	RMSE
1	x	3955.00	0.6583	0.5444	36.310
	x^2^	133.00	0.9885	0.9770	8.155
	x^3^	5.19	0.9996	0.9982	2.279
2	x	4320.00	0.6119	0.4825	37.950
	x^2^	55.82	0.9950	0.9900	5.283
	x^3^	0.59	0.9999	0.9998	0.765
3	x	255.70	0.9487	0.9316	9.232
	x^2^	33.49	0.9933	0.9866	4.092
	x^3^	0.05	1.0000	1.0000	0.214

**Table 3 biomimetics-07-00181-t003:** Fitting results of the spine curves of eight movements.

	Group	1		2		3	
Movements		SSE	Adjusted-R^2^	SSE	Adjusted-R^2^	SSE	Adjusted-R^2^
①		292.5584	0.9950	373.9871	0.9884	33.8155	0.9984
②		995.2212	0.9888	502.3968	0.9881	244.2641	0.9927
③		551.2280	0.9946	444.4963	0.9909	352.6374	0.9924
④		374.7172	0.9965	468.5899	0.9907	352.1559	0.9927
⑤		387.7880	0.9967	801.4724	0.9854	330.8916	0.9956
⑥		346.9201	0.9971	733.1521	0.9866	304.8748	0.9963
⑦		159.6307	0.9988	196.8156	0.9977	311.9123	0.9975
⑧		132.3038	0.9991	184.2575	0.9987	291.8196	0.9981

## Data Availability

Not applicable.

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
