# Peer review of "Analysis of Kinematic Characteristics of Saanen Goat Spine under Multi-Slope"

_biomimetics, 2022, doi:10.3390/biomimetics7040181_

Round 1

Reviewer 1 Report

Ø  The authors address in a satisfactory way the gap of this issue at the introduction section but they should clearly refer the aim of their study at the last paragraph.

Ø  The methodology design (experimental tests, measurements etc) is clear and is reported in details.

Ø  The results of the study are presented in a satisfactory way

Ø  The authors should present the main findings of their study at the first paragraph of the discussion section and also compare their findings to more studies of the literature. Discussion section is to poor

Ø  The authors should add the limitations of their study and propose for further research is mentioned correctly

Author Response

Thank you for your review, please see the attachment.

Reviewer 2 Report

In this paper, the properties of the spine of a goat is analysed under varying conditions with a new tracking system. The paper is well written and results well presented and a complete analysis is given. However, for me, the intrinsic research value of this paper is unclear. Consequently, the paper can be improved by giving additional context and motivation for this case. More specifically, is the value medical, fundamental (can we find answer to a biological hypothesis), engineering (for example what can roboticist learn from this), etc.

Figure 3 and 8: unclear what sampling rate was used to obtained these figures (in the plot a continuous line is plotted). The description on lines 131-136 mentions multiple sampling rates.

Author Response

(The authors gave the same response as above.)

Reviewer 3 Report

General concerns: 

- While the general aim of the study is well chosen, the content is overall hard to grasp and a resulting movement model is lacking.

- Fixing the head of the goat (shown in Fig 1b) might alter normal movements, Please discribe this fact and argue why this has no impact on your findings.

- Section 3.1 is basically unreadable and holds a description of measures. I would expect having setup a model (e.g. via opensim) to model the movement based on this data. Here a generalisation and suitable 3D visualisation of the movements is required. The detailed measures should be moved to supplements.

- The discussion misses a comparison with the findings of sheeps and other studies on the goat. The discussion lacks the corresponding insights for developmend of future tetrapods with flexible bending biomimetic spine structure but remains on a descriptive stage. 

- The visualisation of the findings is challenging. May I suggest using a 3D Visualisation where the range of motion for each marker is visualised and clustered? E.g. one such image per slope.

Minor Comments:

- Section "Therefore, the design of multidirectional flexible ": Here the readers might benefit from a more quantitative discussion of the impact of a "flexible bending biomimetic spine structure". Thus, please add details of the effect the inclusion showed regarding stability measures.

-"The SQbot, a quadruped robot with bionic spinal joints, was also developed." -> Please clarify the context of this sentence.

-"In order to explore the structural characteristics of the sheep spine, " -> Here you suddenly switch from goat to sheep. The comparability among both animals should be discussed in a previous sentence.

- "DeVries et al. [20] used the 6-DF test device to test the flexion and extension, lateral bending and axial rotation of the cervical vertebrae of adult sheep (C2-C7). " -> Should it be "a 6-DF"; here was it placed?

-"However, few studies involve the motion characteristics of the goat spine during movement." -> Name them and discuss them lateron ; Describe what insights are thus missing.

-Section 3.1: Please define what a time 0.0 in the figures represent (healstrike of which foot?) Furtheron discuss the findings based on the gaitcycle events, which should be previously introduced.

Author Response

(The authors gave the same response as above.)

Reviewer 4 Report

The PDF provided contains correction remarks from MS Word. For my review, I supposed that all corrections were accepted.

Nice paper. The impact of the paper could have been improved by providing the perspective of zoology (comparative, functional morphology, evolutionary biology) in the interpretation. At present, the (strong) analysis of one species just gives the base to mirror „goat-like“ locomotion in machines, neglecting the general principles of quadruped mammalian locomotion. Nevertheless, very interesting information for the community.

The only thing I doubt is the cited accuracy of kinematic measurements. Usually, even with human beings (no fur) you are lucky to get residuals near to 1 mm by systems like Qualisys or Vicon, and the soft tissues effects on skin displacement relative to the skeleton are even higher. Thus, I would like to ask you for a discussion on errors and error propagation, and their impact on the accuracy and correctness of the calculated data.

Author Response

(The authors gave the same response as above.)

Round 2

Reviewer 1 Report

The authors improved the manuscript according to reviewers' comments in a satisfactory way.

Author Response

(The authors gave the same response as above.)
